# RGBD Salient Object Detection, Based on Specific Object Imaging

**DOI:** 10.3390/s22228973

**Published:** 2022-11-19

**Authors:** Xiaolian Liao, Jun Li, Leyi Li, Caoxi Shangguan, Shaoyan Huang

**Affiliations:** 1School of Physics and Telecommunications Engineering, South China Normal University, Guangzhou 510006, China; 2School of Electronics and Information Engineering, South China Normal University, Foshan 528225, China

**Keywords:** RGBD salient object detection, specific object imaging, convolutional neural network

## Abstract

RGBD salient object detection, based on the convolutional neural network, has achieved rapid development in recent years. However, existing models often focus on detecting salient object edges, instead of objects. Importantly, detecting objects can more intuitively display the complete information of the detection target. To take care of this issue, we propose a RGBD salient object detection method, based on specific object imaging, which can quickly capture and process important information on object features, and effectively screen out the salient objects in the scene. The screened target objects include not only the edge of the object, but also the complete feature information of the object, which realizes the detection and imaging of the salient objects. We conduct experiments on benchmark datasets and validate with two common metrics, and the results show that our method reduces the error by 0.003 and 0.201 (MAE) on D3Net and JLDCF, respectively. In addition, our method can still achieve a very good detection and imaging performance in the case of the greatly reduced training data.

## 1. Introduction

Inspired by the behavior of the early primate visual system, the visual salient system can rapidly select the location of salient objects that require a detailed analysis in a computationally efficient manner [1]. Salient object detection (SOD) aims to recognize notable objects of a given scene [2]. It is broadly utilized in medical image segmentation [3,4,5], object tracking [6,7], personal re-identification [8,9], monitoring applications [10,11,12], video segmentation [13], etc.

Recently, depth maps can be immediately acquired by devices, such as depth sensors, and the utilization of depth maps can provide the spatial information that is absent from RGB images [14]. RGBD salient object detection (RGBD SOD) has gradually become a research hotspot [15,16,17,18,19]. Lang et al. proposed early on, to use the Gaussian mixture model to discuss the RGBD salient object detection [20]; Peng et al. proposed to use a multi-stage algorithm to fuse depth information and appearance information, to achieve salient object detection, and published the benchmark datasets NLPR [21]. With the development of deep learning, people used neural networks to achieve better results in this field [22,23,24,25,26,27,28]. Chen et al. utilized a multi-modal fusion network to fuse the RGB information and depth information [29]. Fan et al. proposed to use a depth map screening module to filter low-quality depth maps [30]; Fu et al. used a joint learning and dense collaboration module to improve the salient object detection accuracy [31]. At present, all salient object detection methods only transform the object detection problem into an image segmentation problem, and the detection result only contains the outer contour of the salient object [32]. They usually emphasize the edges of the salient objects rather than regions/objects [33]. Annotating only the outer contours of the significant objects can create the risk of misidentification. Because there is uncertainty about natural or man-made objects in the real world. Objects of similar contours may appear in the full-scene image, which is prone to misjudgment, if only the external contour is learned. Different from the above methods, we accept that the genuine detection results should not just contain the outer contour of the target, but need to detect and display the inner detailed information of the target completely. Li et al. proposed a method of imaging objects of interest, which removes the irrelevant background and other objects in the scene, as a label image through innovative processing so that only the image features contained in the target object of the whole scene are learned in the neural network, and finally, only objects of interest are extracted, according to the feature information [34,35].

To settle the above inadequacies of salient object detection, we consolidate the idea of object imaging with a salient object detection method and propose a RGBD salient object detection method, based on specific object imaging. Datasets are an important part of deep learning and affect the accuracy of neural network outputs [36]. Our method adopts a different network learning model from the general RGBD salient object detection methods and redefines the nonlinear model of the network approximation, through input data and output data pairs. We take advantage of the idea that detecting salient objects is the actual direct separation of the salient objects from the irrelevant backgrounds, changing the ground truth to include the full information of the target objects, as shown in Figure 1. The model finally images only specific objects in the scene, while the other information is processed and removed as interference information. So, the detection result can not only realize the detection of the salient target, but also image the target at the same time. The detection result using this method is the complete target after filtering out the irrelevant background, not just the outer contour of the target. To verify the effectiveness of this method, we perform a validation on the SIP datasets [30]. The experimental results show that our method achieves the above advantages and incredibly improves the accuracy of the object detection. The contributions from this paper are summarized in the following.

This paper proposes a salient object detection method, based on specific object imaging, to detect and image salient objects.The experiments are carried out under the benchmark SIP datasets. The proposed new method can also complete the target detection and imaging under the condition of using a small sample dataset. The amount of data required is greatly reduced, and the results are better.

## 2. Related Work

**Traditional RGBD Salient Object Detection.** In early salient object detection work, salient objects were detected from complex scenes, mainly by extracting the handcrafted features of the images. The work has also made substantial progress over the past few decades. Cheng et al. proposed to combine the color contrast, spatial bias, and depth contrast, to provide saliency cues [37]. Ciptadi et al. used an extended disparity framework to model the joint interaction between the depth channels and color information [15]. Cong et al., propose a new metric for evaluating the reliability of depth maps to reduce the impact of low-quality depth maps on salient object detection results [38]. Zhu et al. proposed to use the central dark channel prior to improving the SOD performance [39]. Although some progress has been made in detecting salient objects using traditional methods, based on handcrafted features, their detection results are still sub-optimal. In recent years, SOD methods, based on deep learning, have gradually emerged and demonstrated their advantages.

**RGBD Salient Object Detection, Based on Deep Learning.** Due to the limited capability of the traditional methods, their detection results are still flawed. In recent years, with the rapid development of deep learning, work that combines deep learning with salient object detection has also emerged. Qu et al. first proposed to incorporate CNN into the RGBD saliency object detection, using low-level saliency information to effectively locate the saliency regions in RGBD images [40]. Chen et al., proposed to adaptively select complementary features from different modalities in each feature layer, and then perform a cross-modal cross-feature layer combination [41]. Jiang et al. aimed to learn the optimal view-invariant and consistent pixel-level representations for RGB images and deep images, using an adversarial learning framework [42]. Deep learning-based models can learn higher-level feature representations, and better associate RGB images with depth information, thereby improving the SOD performance.

**Attention Mechanism.** The salient object detection results are easily affected by background clutter. To overcome this problem, many methods now introduce attention mechanisms to improve the detection accuracy. Zhou et al., introduced attention maps to distinguish salient objects from background regions, to reduce the impact of low-quality depth maps on detection results [43]. Li et al., proposed to jointly extract and hierarchically fuse the complementary features of RGB-D images in a dense and intertwined manner. At the same time, an attention mechanism is introduced to locate the potential salient regions in an attention-weighted manner, thereby making progress in highlighting objects and suppressing cluttered background regions [44].

**Specific Object Imaging Method.** Using the idea of an attention mechanism, Li et al. proposed a specific object imaging method [34,35]. This method aims to remove the irrelevant background in the image. By creating a specific dataset label, the neural network only learns the specific target information, and finally realizes the complete imaging of a specific object. The method is optimized, using an end-to-end neural network, which can learn from different training data pairs, each data pair must have a full-scene image $M_{h}$ and a specific object label image $N_{h}$, where *h* = 1, 2,…, *H*. The training process is similar to the optimization process and can be expressed as:(1)$$\left\{\xi ,\theta \right\}=\underset{\xi ,\theta }{\mathrm{argmin}}\frac{1}{H}\sum_{h=1}^{H}\|N_{h}-{\tilde{N}}_{h}\|$$
(2)$${\tilde{N}}_{h}=P_{\theta }\left(O_{\xi }\left(M_{h}\right)\right)$$

The training image $M_{h}$ is passed through the compression network $O_{\xi }$ and the reconstruction network $P_{\theta }$ to obtain the target image $N_{h}$, where ξ and θ are the weights of the compression network and the reconstruction network, respectively, ‖.‖ is the loss function that computes the error between $N_{h}$ and ${\tilde{N}}_{h}$.

This method proposes a visual attention mechanism in a new form, to simulate the processing of cluttered information by biological vision, and this method can provide a new idea for the current salient target detection field, and can also be widely used in other application scenarios of computer vision.

## 3. RGBD Salient Object Detection, Based on Specific Object Imaging

We report an RGBD salient object detection method, based on specific object imaging, whose detection results directly image the inner detailed information of the salient object, while including the outer contour of the object, namely the RGBD salient object imaging. The method uses a neural network model to learn the features of specific objects for achieving the RGBD salient object imaging. In this way, when imaging a complex scene, a specific target object can be directly imaged, and other interfering objects can be filtered out, thereby realizing the active imaging of the specific target object. Its principle diagram is shown in Figure 2. First, we need to collect a certain number of original scene images, and then perform preprocessing to filter out the available images containing specific salient targets, and transfer the available images to the general RGBD network. The training principle is as follows: Suppose a complex scene contains N objects, and the corresponding objects are represented by $k_{i}$ (i = 0, 1, …, N − 1). Then the whole full scene image k can be expressed as:(3)$$k=\sum_{i=0}^{N-1}k_{i},$$

Each object $k_{i}$ can be represented as a linear combination in an M-dimensional sparse space, as shown in the formula:(4)$$k_{i}=\sum_{j=1}^{M}\psi_{ij}x_{ij},$$

Among them, *ψ* is the sparse basis in the sparse space, and *x* is the sparse coefficient in the sparse space. According to the formula, the whole full scene image *k* can be expressed as:(5)$$\;k=\sum_{i=0}^{N-1}k_{i}=\sum_{i=0}^{N-1}\sum_{j=1}^{M}\psi_{ij}x_{ij},$$

Assuming that the salient target in the full scene image *k* is $k_{0}$, the neural network model is used to train the sample image containing the target object $k_{0}$ and its corresponding ground truth to obtain all of the feature information of $k_{0}$. Then, the network model parameters of the trained $k_{0}$ feature are used as the feature filter Φ of the specific target, and the full-scene image *k* is sparsely transformed to the salient object estimated value, as shown in the formula:(6)$$\Phi k=\sum_{i=0}^{N-1}\sum_{j=1}^{M}\Phi \psi_{ij}x_{ij}=\sum_{j=1}^{M}\psi_{0j}x_{0j}={\hat{k}}_{0},$$

Since the feature filter Φ contains the main feature information of the target object $k_{0}$ through the neural network training, it can match the sparse coefficient $x_{0}$ in the feature space $\psi_{0},$ corresponding to the target object $k_{0}$ when performing a sparse transformation on the full-scene image *k*, therefore the signal sparse components of the target object $k_{0}$ is obtained. In this way, the information containing only the specific target object $k_{0}$ is obtained, and the final reconstructed output image $k_{0}$ has only obtained the specific target $k_{0}$.

During the training process of the neural network, a mapping relationship is formed between the input and output of the datasets, and the changed ground truth will change the learning process of the neural network, and finally lead to different results. In other words, the mathematical model defined from the trained NN is completely decided upon by the input and output of the datasets, and they are the data-driven models. The network training process of this method is shown in Figure 3. We pass the full scene image and its corresponding depth map and the specific object label map into the general RGBD network for training. The RGBD network adopts the method of reconstruction after compression to extract the salient objects in the complex scenes. The compression part of the network is used to obtain the characteristics of the salient target, and then the reconstruction part of the network is used to reconstruct the target image. Its formula can be expressed as:(7)$${\hat{S}}_{i}=M_{m}\left(N_{n}\left(k_{i},l_{i}\right)\right),$$
where $k_{i}$ is the input RGB image, $l_{i}$ is the input depth image, $N_{n}$ is the compression sampling network, $M_{m}$ is the reconstruction network, and the obtained ${\hat{S}}_{i}$ is the target image reconstructed by the RGBD network.

In the neural network training step, the compression network and the reconstruction network learn the labels of the different training images, and each pair consists of a label image $S_{i}$ and its corresponding training images $k_{i}$ and $l_{i}$. Since the training process is similar to the optimization process, *m* and *n*, in the above formula, are the weights obtained by the optimization of the compression network and the reconstruction network, respectively, which can be expressed as:(8)$$\left\{m,n\right\}=\underset{m,n}{argmin}\|S_{i}-{\hat{S}}_{i}\|,$$
where ‖.‖ is the loss function that computes the error between the label image $S_{i}$ and the reconstructed image ${\hat{S}}_{i}$.

It is worth noting that the different content of the ground truth in the datasets is extremely important to realize the function of the salient target imaging. With an entirely new ground truth, our new method restricts the regions learned by the neural network of the specific salient targets, which greatly reduces the processing of the amount of data, while also avoiding the learning of irrelevant backgrounds and the analysis and processing of the global scene information.

## 4. Experiments

### 4.1. Datasets

The experiments are performed on the public RGBD benchmark datasets, using the SIP datasets [30]. The datasets contain 929 samples, each of which includes an RGB image and its corresponding depth image, with a resolution of 744 × 992. The main salient object is people, and the background includes some man-made or natural objects, such as buildings, flowers, and vehicles. Among them, only a part of the training set in the SIP datasets participated in the training of the network, the input image is the full-scene image in the SIP datasets, and the output image is the ground truth in the training set. For the salient object detection, based on the specific object imaging, changes in the ground truth will cause differences in the detection results. In order to compare the results presented in the different methods for this dataset, we employ the same parameters as in the original D3NET [30] and JLDCF [31] papers during the training.

### 4.2. Experimental Setup

The hardware of the computer workstation for our network model training is configured as an Intel Core i5-7500 CPU, NVIDIA GeForce GTX 1080Ti GPU. Our method compares with two state-of-the-art salient object detection methods, including D3Net [30] and JLDCF [31]. For fair comparisons, we follow the same model and training settings described in [30,31]. The parameters, such as the loss function, epoch, batch size, optimizer, and momentum are consistent with the source paper, when we compare the two methods. The resolution of the input image is 992 × 744.

### 4.3. Result

We compare the salient object detection methods and the object-specific imaging-based salient object detection methods in D3NET and JLDCF. The experiment uses all 929 standard SIP datasets, of which 870 are used for training and 59 are used for testing. The results of both are shown in Figure 4, and it is evident that the specific object imaging method is more accurate, in terms of detail.

Due to the use of specific object imaging methods, the network learned enough feature information of the significant objects. Therefore, when the dataset decreases, the salient object detection, based on the specific object imaging also shows its superiority under the small sample datasets. We use 200 images in the standard SIP datasets for testing, of which 180 images are used for training and 20 images are used for testing. From the detection results in Figure 5, it can be seen that the general salient target detection method in the small sample dataset will have errors when detecting the edge of the object, and the small edges, such as fingers and clothes, cannot be completely detected. These parts can be completely detected after the object imaging method. Under the condition of small sample datasets, our method can, not only image the target, but also perform excellently in the detection.

### 4.4. Comparison

For evaluation, we adopt the metrics currently widely used in the RGBD salient object detection, namely MAE(M) [45] and S-measure ($S_{\alpha }$) [46]. MAE evaluates the mean absolute error between the salient object detection result S and the ground truth G, and both are normalized to [0, 1], and its basic definition is as follows:(9)$$\mathrm{MAE}=\frac{1}{W\times H}\sum_{x=1}^{W}\sum_{y=1}^{H}\left|\;\hat{S}\left(x,y\right)-\;\hat{G}\left(x,y\right)\right|,$$
where *W* and *H* represent the width and height of the image, respectively.

S-measure pays attention to the structural similarity evaluation that is ignored by MAE, and combines the region-awareness (S_r_) and object-awareness (S_o_) as the definition of the structural similarity evaluation, to obtain a more comprehensive evaluation result. Its definition is as follows:(10)$$S={\alpha *S}_{o}+\left(1-\alpha \right){*S}_{r},$$
where α is set to 0.5 and α ∈ [0,1].

In order to verify the fairness of the experiment, we output the experimental results, based on the specific object imaging method, in the form of a single channel, and compare the results of the two, in the form of a binary. We evaluate the above experimental results, and the quantitative results obtained are shown in Figure 6.

It can be seen from the evaluation indicators that the detection performance is significantly improved when the object imaging method is applied to the salient target detection. In comparison with D3Net, the object imaging SOD method reduces the error of the SOD method by 0.003 (the evaluation metric is MAE). In comparison with JLDCF, the object imaging SOD method improves the structural similarity by 0.061 (the evaluation metric is the S-measure). This demonstrates the effectiveness of the salient object imaging. Under the small amount of data, our new method can, not only detect the correct target effectively, but also image the target. It ensures that the feature information about the target is preserved while detecting the target.

## 5. Conclusions

We propose a RGBD salient object detection method, based on specific object imaging, which images the object while achieving the salient object detection. We conducted experiments on the benchmark datasets. The experimental results show that the accuracy of the salient object detection, based on the object imaging, is higher than that of the general salient object detection. This method can not only detect the target but also retain the characteristic information of the target to further improve the recognition of the target. This method has the same applicability to other networks, so this method is expected to open a new road in the application of general image processing. However, at present, the method we propose can only deal with similar objects in the scene. In the future, we will adopt the idea of the parallel network, to solve the situation of detecting and imaging multiple types of objects in the scene.

## Figures and Tables

**Figure 1 sensors-22-08973-f001:**
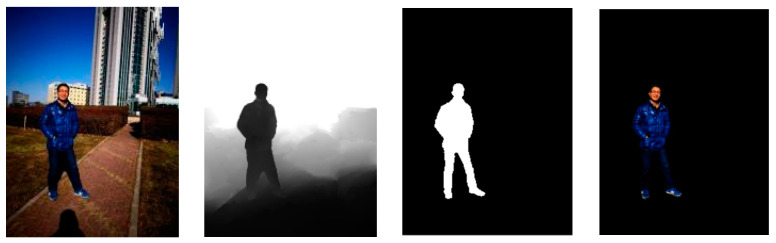
From left to right, RGB images, depth images, the ground truth of the SOD method, and the ground truth of our method.

**Figure 2 sensors-22-08973-f002:**
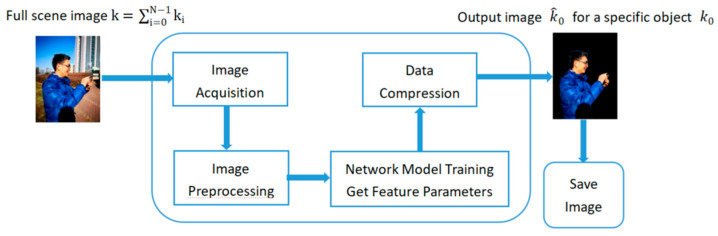
Schematic diagram of a specific object imaging. First of all, the full-scene image is obtained, and then the preprocessing of image screening is sent to the neural network for training, and to filter out the irrelevant background by feature information of a specific object obtained by training. Finally, the data is compressed and reconstructed to get our target image.

**Figure 3 sensors-22-08973-f003:**
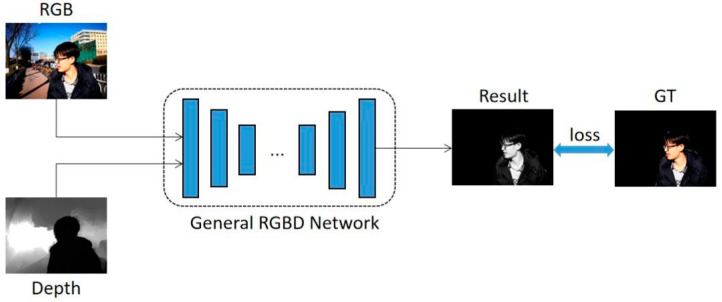
The neural network training process for the RGBD salient object detection, based on the specific object imaging. Following the input of the RGB map and the depth map into the general RGBD network, it is trained with the label map of the specific object imaging method, and finally the detection and imaging results containing only the target are obtained.

**Figure 4 sensors-22-08973-f004:**
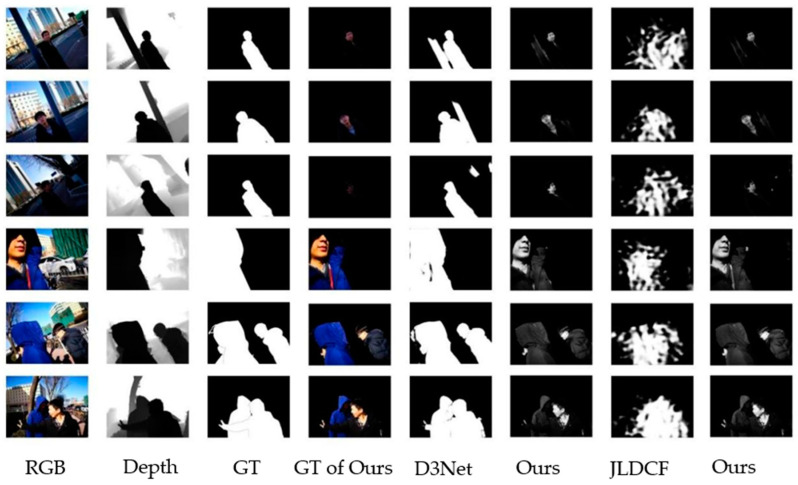
Visual comparison of our method with the traditional salient object detection methods on D3Net and JLDCF.

**Figure 5 sensors-22-08973-f005:**
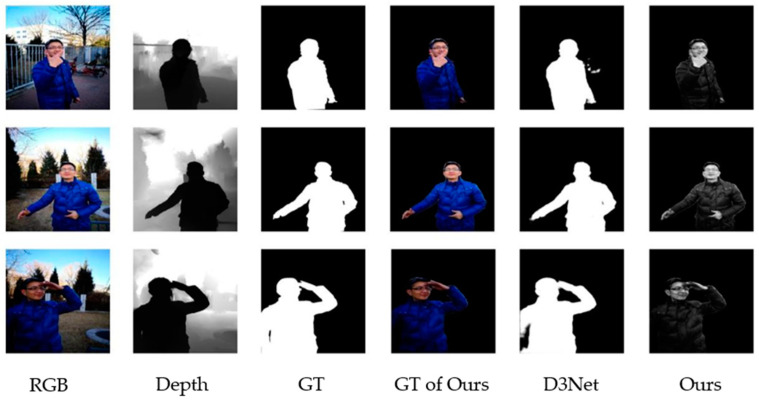
Visual comparison of the SOD methods and the SOD methods, based on specific object imaging on small sample datasets.

**Figure 6 sensors-22-08973-f006:**
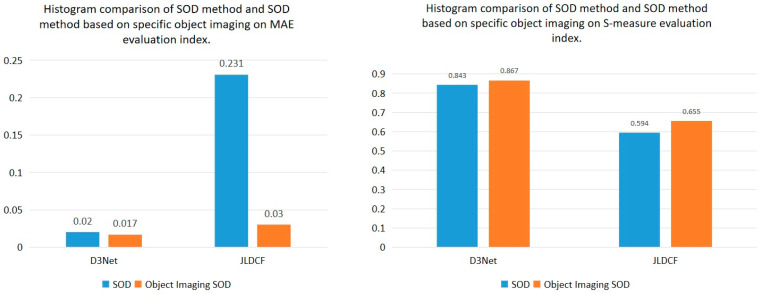
Comparison of the histograms of the SOD method and the SOD method, based on specific object imaging in the two evaluation indicators.

## Data Availability

Not applicable.

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
