# Peer review of "RGBD Salient Object Detection, Based on Specific Object Imaging"

_sensors, 2022, doi:10.3390/s22228973_

Round 1

Reviewer 1 Report

In the paper “RGBD Salient Object Detection Based on Specific Object Imaging” the authors propose a method for improving the quality of models for problems of detection of salient objects. The approach is based on a deep learning model that performs image segmentation based on two inputs - RGB and depth images. The main idea of the method is to modify training labels by replacing the object binary mask with its content from the RGB input image. The main problem of the article is the weakly justified method and why it can leads to an increase in quality.

First of all, it is not clear why proposed GT labels are better than standard SOD binary masks. Learning with binary images as answers seems much easier and well-defined. This should be described in more detail. 

Secondly, comparison with JLDCF seems to be inconsistent - provided results in figure 5 for JLDCF method looks like it was performed with some mistakes. It is very strange that a model trained on 870 images performs so poorly. In addition, it is unclear why the proposed training method requires less data and there is no explanation in paper.

Further, it is also hard to compare results of the proposed method with D3Net in figure 4 due to different output formats: color object with many dark parts on black background vs binary mask. For a more convincing comparison it would be better to convert the result of your model to binary form. Comparison section with metrics should describe more detailed how metrics were applied, how the various outputs were converted into a single format for comparison, and so on. Figure 6 looks uninformative since the same graph shows values ​​with different interpretations (lower is better for MAE and vice versa for S-measure).

Finally, this paper requires more experimentation. Provided experiments seems weak: the first one with D3Net was run on a very small test set of 20 images, mostly with the same people. The comparison with JLDCF does not show anything since the model was poorly trained. The article also does not contain any details regarding the neural network model used, its architecture and loss function. It also does not say how many times the training of the model was reproduced and whether the result is random.

The paper can not be accepted in its current form and needs major corrections.

Reviewer 2 Report

This paper presents a RGBD salient object detection method based on specific object imaging. There are some concerns. The authors need to refine this paper before it can be accepted.

- The contributions should be further discussed and explained. In addition, it is also clear what challenge the claimed contributions can address.

- It is better to cite more papers about the saliency detection, e.g.: Industrial pervasive edge computing-based intelligence iot for surveillance saliency detection, ieee tii; Trustful internet of surveillance things based on deeply-represented visual co-saliency detection, ieee iot; Salient object detection in the distributed cloud-edge intelligent network, ieee network. The authors could list more studies by themselves.

- Please improve the image resolution of the figures.

- Please enrich the captions of figures and tables for better clarification.

- For Figure 6, it is suggested to remove the Excel style of this figure.

- In Figure 4, it is confused that why there are two GT ? It is common that just one GT.

- Some grammatical errors and typos.

Reviewer 3 Report

The authors presented RGBD Salient Object Detection Based on Specific Object Imaging. All the experiments were carried out using the public dataset and compared with the two state-of-art methods. The article scientifically sounds good, but, needs to improve the methods and explain the relevancy of the article. The Major comments:

1. The methodology is not well explained. Authors only mention theoretical information but are not able to explain what methods, algorithms, etc., are used for all the experiments.  For example, the authors mentioned in Figure 2 that the pre-processing was carried out before feeding the Neural Network but never explain what type of image pre-processing was carried out. Besides that, the proposed neural network is also not presented properly. Improper or insufficient methods mislead the results and the results can not be trusted. I would suggest to re-write sections 3 and section 4 with the proper graphical information and proper explanation. In the experiments, the authors used different parameters to compare the results, but never explain why did they choose different training parameters. 

2. The presentation of the results is not sufficient. I would suggest to re-write results and discussion section by considering the following points:

* Add one or two sections for results and discussion.

*The authors need to prove by comparing the results with other state-of-art articles, that this research study is scientifically sound. 

Minor comments:

* Abstract, Line-10, should be small e, not E.

* The last sentence of the abstract is not clear, please re-write it.

* Introduction, Line28, .....other fields, what are those other fields?

* Figure 2, does not explain the steps of the block diagram in the caption. Please double-check the caption formatting. 

* The numbering is wrong. Experiments should Section 4.

Round 2

Reviewer 1 Report

In the paper “RGBD Salient Object Detection Based on Specific Object Imaging” the authors propose a method for improving the quality of models for problems of detection of salient objects. The approach is based on a deep learning model that performs image segmentation based on two inputs - RGB and depth images. The main idea of the method is to modify training labels by replacing the object binary mask with its content from the RGB input image. The main problem of the article is the weakly justified method and why it leads to an increase in quality.

First of all, it is not clear why proposed GT labels are better than standard SOD binary masks. Learning with binary images as answers seems much easier and well-defined. This should be described in more detail. 

Secondly, comparison with JLDCF seems to be inconsistent - provided results in figure 4 for JLDCF method looks like it was performed with some mistakes. It is very strange that a model trained on 870 images performs so poorly. In addition, it is unclear why the proposed training method requires less data and there is no explanation in paper.

Further, it is also hard to compare results of the proposed method with D3Net in figures 4 and 5 due to different output formats: color object with many dark parts on black background vs binary mask. For a more convincing comparison it would be better to convert the result of your model to binary form. 

Finally, this paper requires more experimentation. Provided experiments seems weak: the first one with D3Net was run on a very small test set, mostly with the same people. The comparison with JLDCF does not show anything since the model was poorly trained. The article also does not contain any details regarding the neural network model used, its architecture and loss function. It also does not say how many times the training of the model was reproduced and whether the result is random.

Author Response

Please check the attached file:

Reviewer 2 Report

No further question.

Author Response

Dear reviewer,

Thank you so much for your reviewing! We deeply appreciate your recognition of our research work.

Best wishes,

Jun Li

Reviewer 3 Report

The paper is improved enough to be published. 

Author Response

(The authors gave the same response as above.)
